# Cerebrospinal Fluid Protein Concentration in Healthy Older Japanese Volunteers

**DOI:** 10.3390/ijerph18168683

**Published:** 2021-08-17

**Authors:** Tatsuya Yoshihara, Masayoshi Zaitsu, Kazuya Ito, Ryuzo Hanada, Eunhee Chung, Rie Yazawa, Yukikuni Sakata, Koki Furusho, Hiroshi Tsukikawa, Takeshi Chiyoda, Shunji Matsuki, Shin Irie

**Affiliations:** 1SOUSEIKAI Fukuoka Mirai Hospital Clinical Research Center, Kashiiteriha 3-5-1, Higashi-ku, Fukuoka 813-0017, Japan; koki-furusho@lta-med.com (K.F.); hiroshi-tsukikawa@lta-med.com (H.T.); shunji-matsuki@lta-med.com (S.M.); shin-irie@lta-med.com (S.I.); 2Department of Public Health, Dokkyo Medical University School of Medicine, 880 Kitakobayashi, Mibu-machi, Shimotsuga-gun, Tochigi 321-0293, Japan; m-zaitsu@dokkyomed.ac.jp; 3SOUSEIKAI Clinical Epidemiological Research Center, Kashiiteriha 3-5-1, Higashi-ku, Fukuoka 813-0017, Japan; kazuya-ito@lta-med.com; 4College of Healthcare Management, Takayanagi 960-4, Setaka-machi, Miyama 835-0018, Japan; 5SOUSEIKAI Sumida Hospital, 1-29-1, Honjo, Sumida-ku, Tokyo 130-0004, Japan; ryuzo-hanada@lta-med.com (R.H.); rie-yazawa@lta-med.com (R.Y.); yukikuni-sakata@lta-med.com (Y.S.); takeshi-chiyoda@lta-med.com (T.C.); 6SOUSEIKAI Global Clinical Research Center, Kashiiteriha 3-5-1, Higashi-ku, Fukuoka 813-0017, Japan; eunhee-chung@lta-med.com

**Keywords:** cerebrospinal fluid, total protein, reference interval, older population, Japanese healthy volunteer

## Abstract

The concentration of cerebrospinal fluid total protein (CSF-TP) is important for the diagnosis of neurological emergencies. Recently, some Western studies have shown that the current upper reference limit of CSF-TP is quite low for older patients. However, little is reported about the concentration of CSF-TP in the older Asian population. In this study, we retrospectively analyzed the CSF-TP concentrations in healthy older Japanese volunteers. CSF samples in 69 healthy Japanese volunteers (age range: 55–73 years) were collected by lumbar puncture, and the data of CSF were retrospectively analyzed. The mean (standard deviation) CSF-TP was 41.7 (12.3) mg/dL. The older group (≥65 years old) had higher CSF-TP concentration than the younger group (55–64 years old). The 2.5th percentile and 97.5th percentile of CSF-TP were estimated as 22.5 and 73.2 mg/dL, respectively, which were higher than the current reference range in Japan (10–40 mg/dL). Conclusions: The reference interval of CSF-TP in the older population should be reconsidered for the precise diagnosis of neurological emergencies.

## 1. Introduction

Worldwide, the concentration of cerebrospinal fluid total protein (CSF-TP) is used for the diagnosis of various life-threatening neurological emergencies, such as infectious, inflammatory, and immune diseases of the central/peripheral nervous systems; thus, a valid reference interval of CSF-TP concentrations is crucial for precise diagnosis and treatment.

The current gold standard, the reference range of 15–45 mg/dL (0.15–0.45 g/L) for lumbar CSF-TP dates back to 1938, and it has been adopted for a wide range of population for almost a century [1,2]. However, recent Western studies have started to report that the concentration of CSF-TP in adults is considered to increase gradually with age, and the concentrations in the older population are significantly higher than the current standard upper reference limit [1,3,4,5]. Consequently, several reports have recommended to update the threshold to reflect the actual CSF-TP concentration, especially in the older population [1,3,4,6,7,8,9]. Meanwhile, a lower reference range of 10–40 mg/dL has been widely adopted for all ages in Japan without citations [10]. Little has been reported about the concentration of CSF-TP in Asian populations. To our knowledge, only two studies have reported on CSF-TP in Japanese patients, but these were presented over 30 years ago [11,12]. Since the CSF measuring methods and the accuracy of testing have been greatly improved in the past decades, it is crucial to review the CSF-TP reference interval for various populations [2,11,12]. In addition, even though the available data from a healthy population are often limited [1,3,6,13] because of the invasive nature and the potential risks of the sampling procedure, using the data of healthy subjects is desirable to properly interpret the laboratory data [14].

Here, we revisit the CSF-TP levels in a healthy older population in Japan. In our facilities, CSF sampling in healthy volunteers has been performed as part of clinical trials. Therefore, we retrospectively analyzed the clinical laboratory data of CSF in healthy older Japanese volunteers and assessed whether the CSF-TP level linearly increased with age.

## 2. Materials and Methods

### 2.1. Study Design and Ethics

This study was approved by SOUSEIKAI Hakata Clinic Institutional Review Board (Approval number: M-79). This single-center retrospective study was designed to analyze the de-identified CSF laboratory data from October 2013 to March 2019 at SOUSEIKAI Mirai Hospital Clinical Research Center. The study was conducted in compliance with the Declaration of Helsinki and, Ethical Guidelines for Medical and Health Research Involving Human Subjects, Japan.

### 2.2. Research Subjects and CSF Sampling

All CSF sampling procedures were carried out as a part of four clinical trials from October 2013 to March 2019 at SOUSEIKAI Mirai Hospital Clinical Research Center. Each study protocol was reviewed and approved by SOUSEIKAI Hakata Clinic Institutional Review Board. All CSF samples were collected by a lumbar puncture by experienced in-house anesthesiologists after a thorough explanation of the procedure to the subjects and after written informed consent was given. The collection volumes of CSF samples were 6–12 mL depending on the study protocols. The participants were healthy Japanese adults with no existing neurological symptoms or signs, no abnormal laboratory findings that indicated inflammatory diseases, and no medical treatments including using medical drugs. The total protein concentration, glucose levels, and the cell numbers in the CSF were measured using the pyrogallol red method, glucokinase method, and Fuchs-Rosenthal method, respectively, at a commercial laboratory, LSI Medience Corporation (Tokyo, Japan).

### 2.3. Statistical Analysis

The results are shown as mean ± standard deviation (SD). The difference of CSF-TP concentrations between males and females and the difference between age groups (55–64 years vs. ≥ 65 years) were analyzed by the Wilcoxon rank-sum test. The lower and upper reference limits of the CSF-TP were estimated to be the 2.5th percentile and the 97.5th percentile, respectively [14]. According to Horn’s method of outlier detection [15], one male volunteer was excluded from the analyses because his CSF-TP concentration (104 mg/dL) was more than Q3 (upper quartile) + 1.5 interquartile range after Box-Cox transformation. To assess the association between age and CSF-TP concentrations, regression coefficients (β) and 95% confidence intervals (CI) of CSF-TP concentrations against age were estimated with an ordinary least squares regression model.

Our study population was a relatively older population and did not include the individuals younger than 55. Therefore, in the supplementary analysis, we compared our CSF-TP concentrations to previous published data [3,4,6,11,12,16,17] by *t*-tests. To be used as historical controls, the studies should have included younger study subjects and have data availability regarding the mean CSF-TP concentrations, SD, and the number of subjects.

Two-tailed *p*-values < 0.05 were considered as statistically significant. Statistical analyses were performed using JMP Pro 15 (SAS Institute Inc., Tokyo, Japan) and Graph Pad Prism 9 (MDF Co., Ltd. Tokyo, Japan).

## 3. Results

The data of 69 Japanese healthy older volunteers were analyzed. Thirty-seven (54%) were male. The age range was 55–73 years, and the mean age was 62.8 ± 5.2 (male, 62.2 ± 5.3; female, 63.5 ± 5.1). The mean CSF-TP concentration in all participants was 41.7 ± 12.3 mg/dL, which is higher than the general upper reference limit in Japan (40 mg/dL) (Table 1). The distribution of CSF-TP was right-skewed compared to the normal distribution (Figure 1A), which was consistent with a former report [3]. The 2.5th and 97.5th percentile of CSF-TP values in all participants were estimated at 22.5 mg/dL and 73.2 mg/dL, which were higher than the current lower and upper reference limit of CSF-TP in Japan, respectively (Table 1). There was no sex difference in CSF-TP concentration (Figure 1B).

Although the linear regression analysis shows that the CSF-TP concentrations were not significantly correlated with age [R = 0.21, β (95% CI) = 0.51 (−0.06 to 1.07), *p* = 0.08] (Figure 1C), the older age group (≥65 years old) had a significantly higher CSF-TP concentration than the younger age group (55–64 years old) (Table 2). In the supplementary analysis, the mean CSF-TP concentrations of younger populations appeared to be lower than that of the present study (Table 3).

The mean glucose value in CSF was 59.9 ± 5.5 mg/dL (Table 1), which is within the normal Japanese adult range (50–75 mg/dL). The 2.5th and 97.5th percentile of CSF-glucose values in all participants were estimated at 49.8 mg/dL and 72.0 mg/dL, respectively, which are roughly consistent with the current reference range in Japan (50–75 mg/dL) (Table 1). There was no difference in the values of CSF-glucose between the age groups (Table 2). The cell counts in CSF were conducted in 53 volunteers, and the cell numbers were 0–5/µL, which were within the normal range (0–5/µL).

## 4. Discussion

In the present study, the 2.5th and 97.5th percentiles of CSF-TP concentration in the Japanese older population was higher than the current lower and upper reference limit respectively in Japan, and the older population had a higher CSF-TP concentration than the younger population.

CSF protein concentration is known to increase in various clinical situations such as a tumor, inflammation, bleeding, or injury in the central nervous system. It is also well accepted that the protein concentration in the CSF is correlated with serum protein concentrations, reduced CSF turnover, the blood–brain barrier (BBB), or blood–CSF barrier functions [18,19,20,21]. Fetuses, neonates, and young infants are known to have high CSF-TP concentrations [22,23,24] due to the possible immaturity of the BBB [19], high transcellular transfer of plasma proteins across choroid plexus epithelial cells, and the slow turnover of CSF in the developing brain [25]. In recent years, it has been reported that CST-TP concentration in adults increases with age [1,6,17]. Although the etiology of the higher CSF-TP concentration in the older population has not been well elucidated, some studies have suggested mechanisms, such as reduced CSF turnover, integrity changes in BBB, or lumber stenosis in the older population [8,18,19]. However, the age-specific reference intervals have not been actively investigated due to the invasive nature of the procedure to collect the samples, especially in healthy volunteers.

In this study, to investigate the CSF-TP levels in healthy older volunteers in Japan, we collected and reviewed CSF-TP data from 69 healthy Japanese volunteers aged 55 to 73 years old from clinical study data. From the analysis, we found that both the 2.5th and 97.5th percentiles of CSF-TP concentration in the study population was higher than that of the general lower and upper reference limits in Japan (10 mg/dL and 40 mg/dL, respectively). Forty five percent of our population had higher CSF-TP concentration than the general upper reference limit in Japan. Even using an upper reference limit of 45 mg/dL which has been used worldwide, 31% of the volunteers had higher CSF-TP concentrations, which is consistent with the previously mentioned report from a group of Western older adults [3]. From these results, it is concerning that the current upper reference limits might lead to a false-positive determination that may cause excessive interventions in a fairly large percentage of patients, without resulting in the diagnosis of a neurological disorder [1,7,8,9]. Some studies have suggested that use of age-adjusted upper reference limit for CSF-TP concentration improves the diagnostic accuracy in abnormal neurological statuses, such as albuminocytological dissociation, Guillan-Barré syndrome, or chronic inflammatory demyelinating polyneuropathy [7,8,9]. Moreover, in the clinical trials regarding candidate drugs for diseases of the central nervous system, there are growing needs for the assessment of the biomarkers in CSF to establish the safety profiles in the central nervous system in both young and old healthy volunteers. Thus, it is essential to establish the correct age-specific CSF-TP reference interval for accurate subject safety assessments and precise interpretation of the information obtained from the CSF samples. Breiner et al. proposed that the upper reference limit should be stated age-specifically and that the values over 60 mg/dL should be applied for patients aged 50 and above [1]. Our data may also support the idea that the upper reference limit of CSF-TP should be age-specific.

The concentrations of CSF-TP are known to differ according to the neuroaxis [3,10,13]. According to a laboratory data book in Japan, the reference intervals of CSF-TP are 10–15 mg/dL in the cerebral ventricles, 15–25 mg/dL in the cisterna magna, and 20–40 mg/dL in the lumbar sac [10]. In the present study, the 2.5th percentile of CSF-TP values in our population was estimated at 22.5 mg/dL, which was also consistent with a former report [4] and higher than the general lower reference limit in Japan (10 mg/dL). Our results also suggest that the lower reference limit of CSF-TP in the older population may have to be updated to 20 mg/dL rather than 10 mg/dL when the CSF samples are collected by lumbar puncture.

Males have been reported to have a slightly higher CSF-TP concentration than females [4,16]. Our data also showed that a somewhat higher mean CSF-TP concentration in males than in females. However, the difference is small and not statistically significant; hence, we consider that a specific reference interval for each gender may not be necessary for the older Japanese population.

### Limitations and Strengths of the Study

This study has some limitations. First, the number of volunteers we analyzed was relatively small compared with previous extensive studies [1,3,4,6]. However, the concentrations of CSF-TP in the present study were consistent with the previous studies’ values. Second, the study participants were healthy volunteers (but not patients) in a clinical trial, and we could not follow up them after the trial. Therefore, it remains unclear whether those with high CSF-TP levels subsequently are associated with the future development of neurological symptoms. Third, there is no control cohort with diseases that cause elevated CSF-TP levels since the data of CSF-TP in healthy volunteers were analyzed in this study. Fourth, the volumes of CSF samples we collected varied on each study protocol (6–12 mL). However, although it is known that the concentrations of CSF-TP differ along with the neuroaxis [3,10], the variation of the sampling volumes should not be a concern because this factor is reported not to impact the CSF-TP concentration when the sampling volumes were less than 12 mL [26]. Fifth, our population was 55 years old and above. To determine the reference interval of CSF values for Japanese adults, more extensive studies including subjects aged 54 and younger are needed. However, the supplementary analysis shows that the younger population has lower CSF-TP values than our population.

One of this study’s strengths is that our data are based on the data from healthy volunteers who had no existing neurological symptoms, abnormal laboratory findings, or medical treatments, although neurological diseases had not been excluded by imaging tests such as magnetic resonance imaging. Due to the invasiveness and the potential risks following the CSF sampling, it is challenging to collect CSF samples in healthy volunteers when they have no clinical symptoms. Thus, most reports on CSF data were based on patients who needed to undergo CSF examinations or lumbar anesthesia. Although patients who had neurological diseases had been excluded from the analyses when published [1], there might be a possibility that the patients’ health conditions had affected the CSF-TP concentrations. However, since our results were roughly consistent with the previous Western reports [1,3,4,6], this study will be another supportive data of CSF-TP concentration in the older population.

## 5. Conclusions

The current reference limits of CSF-TP, particularly the upper reference limit, may not be reflecting the physiological conditions of older adults. Re-evaluating the reference interval of CSF-TP for older populations should be considered to increase the precision of diagnosing neurological emergencies, as well as to safely assess clinical study data.

## Figures and Tables

**Figure 1 ijerph-18-08683-f001:**
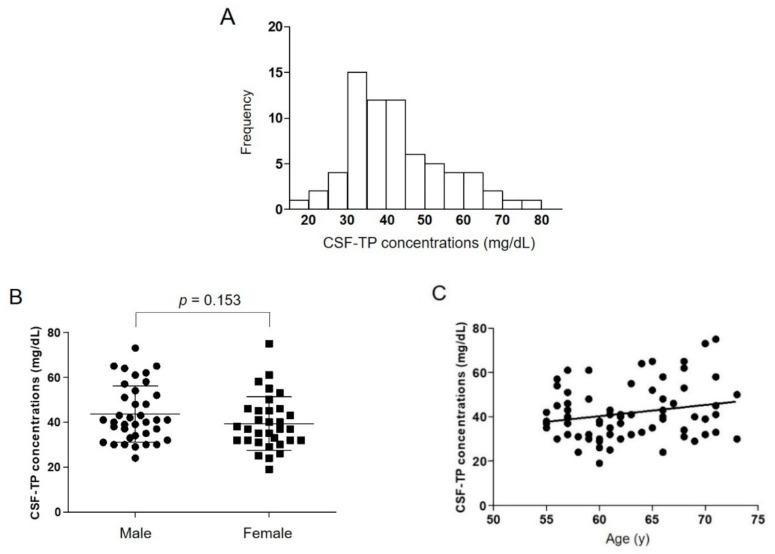
Distribution of cerebrospinal fluid total protein (CSF-TP) concentrations (**A**). CSF-TP concentrations in Japanese healthy older volunteers by sex (**B**). Correlation of age and CSF-TP concentration (**C**).

**Table 1 ijerph-18-08683-t001:** Mean, 2.5th and 97.5th percentiles of CSF-TP and CSF-glucose.

	Mean (SD)	2.5th Percentile	97.5th Percentile
CSF-TP (mg/dL)			
All participants (*n* = 69)	41.7 (12.3)	22.5	73.2
Male (*n* = 37)	43.7 (12.5)	24.4	72.5
Female (*n* = 32)	39.4 (11.9)	19.0	75.0
CSF-glucose (mg/dL)			
All participants (*n* = 69)	59.9 (5.5)	49.8	72.0
Male (*n* = 37)	61.1 (5.1)	49.2	72.8
Female (*n* = 32)	58.6 (5.7)	50.0	72.0

CSF, cerebrospinal fluid; TP, total protein; SD, standard deviation.

**Table 2 ijerph-18-08683-t002:** The concentrations of CSF-TP and CSF-glucose stratified by age.

Age Group	55–64, Year (*n* = 42) Mean (SD)	≥65, Year (*n* = 27) Mean (SD)	*p*-Value ^a^
CSF-TP (mg/dL)	39.0 (10.5)	45.9 (13.9)	0.044
CSF-glucose (mg/dL)	60.3 (6.1)	59.4 (4.5)	0.658

^a^ Wilcoxon rank-sum test; CSF, cerebrospinal fluid; TP, total protein; SD, standard deviation.

**Table 3 ijerph-18-08683-t003:** Comparison of CSF-TP concentration with previous reports.

Historical Controls	Setting	Subject	Mean Age (Range), year	Number	Mean CSF-TP (SD), mg/dL	*p*-Value ^a^
Hirohata et al. 1984 [12]	Japan	Patients with no neurological disease	34.2 (13–57)	30	29.9 (9.0)	<0.001
Breebaart et al. 1978 [16]	Netherlands	Patients with no neurological disease	38	139	31 (18)	<0.001
Takeoka et al. 1976 [11]	Japan	Patients with no neurological disease	41.0 (18–77)	27	30.0 (6.9)	<0.001
Hegen et al. 2016 [3]	Austria	Patients with no neurological disease	42.1 (18–80)	332	40.2 (12.6)	0.367
McCudden et al. 2017 [4]	Canada	Female patients with no neurological disease	43 (18–97)	3804	32 (10)	<0.001
		Male patients with no neurological disease	44 (18–94)	2264	38 (11)	0.006
Atack et al. 1988 [17]	USA	Healthy normal subjects	48.1 (20–86)	26	49 (10)	0.069
Dufour-Rainfray et al. 2013 [6]	France	CSF samples with cellularity and high glucose levels were excluded	(50–100)	1192	48 (24)	0.031

^a^*t*-test compared with the summary statistics of the current study: mean CSF-TP concentration, 41.7 (SD 12.3) mg/dL; SD, standard deviation.

## Data Availability

The data that support the findings of this study are available from the corresponding author, TY, upon reasonable request, according to the Ethical Guidelines for Medical and Health Research Involving Human Subjects, Japan.

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
