# Peer review of "Cerebrospinal Fluid Protein Concentration in Healthy Older Japanese Volunteers"

_ijerph, 2021, doi:10.3390/ijerph18168683_

Round 1

Reviewer 1 Report

This is a simple but well-conducted and interesting study on an important theme. The only weakness in my opinion is the small size of the study cohort. However, I acknowledge that it is difficult to recruit healthy subjects for CSF studies, and the authors have discussed this point.

Reviewer 2 Report

I have read with great interest the manuscript from Yoshihara et al with the title “Cerebrospinal fluid protein concentration in Japanese healthy older volunteers”. I found the study very interesting, nicely presented and with well-supported conclusions. Overall, I have no objections to the manuscript, but there are two points that I think the authors should discuss in the text before proceeding with publication:

  1. The samples were obtained from the patients during a long period of 6 ½ years, with the first ones dating back in October 2013. Have the authors followed up the subjects in terms of their health within these 8 years? Were they able to correlate potentially elevated levels of CSF-TP with health deterioration or the development of neurological symptoms? I think that making a connection to the current or at least the most recent health status of these patients would significantly add to the study and potentially allow the authors to identify subjects that at that point seemed healthy but their slightly higher level of CSF-TP could have indicated the future development of symptoms at a later stage.
  2. Even though the main point of the manuscript is the study of CSF-TP levels in healthy subjects as it is also clearly stated in the title, I find that it is actually missing they “positive control” (if I am allowed to call it this way) of disease cases to demonstrate the elevated levels of CSF-TP. I think this point should be included in Limitations and Strengths of the manuscript.

Round 2

Reviewer 2 Report

Dear editor,

the authors have addressed my concerns adequately both in their response letter and in the revised version of the manuscript. As I mentioned in my initial report, I think the study is very interesting and in my opinion it should go ahead for publication.